# Sustainability Assessment of Constructive Solutions for Urban Spain: A Multi-Objective Combinatorial Optimization Problem

**Simón Martínez [1,\*], Cristina González [1]**, **Antonio Hospitaler [2]** and **Vicente Albero [2]**

[1] Department of Construction and Manufacturing Engineering, Escuela Técnica Superior Ingenieros Industriales, Universidad Nacional Educación a Distancia, 28015 Madrid, Spain; cggaya@ind.unes.es

[2] Concrete Science and Technology Institute, ICITECH, Universitat Politècnica de València, 46022 València, Spain; ahospitaler@cst.upv.es (A.H.); valbero@mes.upv.es (V.A.)

[\*] Correspondence: smartinez200@alumno.uned.es

**Abstract:** Industrial areas are set up on plots of roads and associated infrastructure. These use materials and machinery that have environmental impacts, and thus require constructive solutions throughout their lifecycles. In turn, these solutions and their components cause environmental impacts that can be measured by sustainability indicators. The concept of sustainability is closely tied to sustainable development, which is defined as "development that meets the needs of the present, without compromising the ability of future generations to meet their own needs". The large number of possible and available solutions means that identifying the best one for a given road section must employ a set of heuristic techniques, which conceptualize the issue as a combinatorial optimization problem that is purely discrete and non-differential. The system chosen can be based on a genetic algorithm method that differentiates individuals based on three sustainability indicators: $CO_2$ emissions, embedded energy (also known as embodied energy, defined as the energy expended to manufacture a product), and economic cost. In this paper, we supplement traditional cost analyses using a three-objective multi-objective genetic algorithm that considers the aforementioned criteria, thus addressing sustainability in aggregate planning. The procedure is applied to three objective functions—$CO_2$ emissions, economic cost and embedded energy—for each possible solution. We used the non-dominated sorting genetic algorithm (NSGA-II) to implement multi-objective optimization in MATLAB. Additional results for a random walk and multi-objective search algorithm are shown. This study involved 26 design variables, with different ranks of variation, and the application of the algorithm generates results for the defined Pareto fronts. Our method shows that the optimal approach effectively solves a real-world multi-objective project planning problem, as our solution is one of the Pareto-optimal solutions generated by the NSGA-II.

**Keywords:** industrial parks; eco-design; urban planning; sustainability assessment; genetic algorithms; multi-objective optimization

---

## 1. Introduction

In urban environments, the concept of sustainability refers to the capacity of the urban area to sustain the community's desired quality of life, without causing adverse environmental impacts that would compromise the ability of contemporary and future populations to do the same, both within and outside the municipal limits. That is, models of sustainable urban development imply urban and spatial planning that considers resource use and aims to limit environmental impact [1].

In high-income countries, particularly in Europe, some urban regeneration initiatives have adopted this approach, applying policies to promote density through processes involved in the

"compact city" model [2]. Urban expansion, in contrast, generally results in a high consumption of land, urban congestion, increasing infrastructure costs, and worsening population health, among other negative effects [3].

In addition, many planners consider the construction of green areas unsustainable (particularly when it is possible to regenerate or reuse abandoned land or gray land), as it depletes the non-renewable resources of the land [4–7].

## 1.1. The Eco-Industrial Park, a New Industrial Urban Model

While the aggregation model of traditional industrial districts is primarily based on economic opportunities and accessibility, eco-industrial parks (EIPs) challenge this production model. An EIP is an industrial park, in which businesses cooperate with each other and with the local community in an attempt to share resources; reduce pollution and waste; and search for greater economic benefits, greater environmental quality, and sustainability. EIPs invest considerable effort into managing environmental impacts to achieve the maximum possible efficiency in the use of natural and non-renewable resources (including water, energy, and materials) and to limit emission and waste [8,9].

Historically, urban development was largely a reactionary process within the wave of industrialization in the 19th century. Industrialization profoundly transformed urban and spatial contexts, straining the limits of urban habitability as the population underwent polarization processes. Many planners responded by conceiving new urban models to resolve the conflict between industrial progress and the quality of life of city dwellers. Howard, E., Garnier, T., and Le Corbusier, F. [10] are just a few of the innovators who proposed city planning models characterized by a relatively pronounced divide between residential and industrial functions. These precepts were referential in the urban planning and development that followed across Europe.

This classical urban model pivoted on economic questions related to profits and technological innovation, and the location model was principally concerned with removing industrial complexes from urban centers, while also ensuring access to transport corridors and strategic hubs [11]. This model dominated the urban development landscape until the 1970s, when the environmental consequences of unchecked industrialization crystallized in the evident deterioration of air and water quality.

The EIP model, which appeared in the 90s, responded to these acute challenges with innovative strategies to meet the environmental objectives of cutting emissions and energy and resource use, without undermining economic growth. Early on, Lowe et al. [12] established the idea of an eco-industrial park as "a community of manufacturing and service companies seeking better environmental and economic performance through collaboration in the management of environmental issues and resources, including energy, water and materials".

EIPs have a foundation in scientific disciplines, with strong ties to sustainability, especially industrial ecology, whose tenets include energy efficiency, closed production circuits, and, above all, industrial symbiosis, understood as interlocking systems (as opposed to silos) where materials, energy, and information flow in isolation from each other [13].

Côte and Hall [14] built on this conceptual foundation, defining an EIP as "an industrial system that conserves natural and economic resources, reduces the costs and responsibilities of production, material energy, insurance and treatment, improves operational efficiency and quality, the health of the worker and the public image, and provides opportunities for the generation of income from the use and sale of wasted materials". Roberts [15] also highlighted the interest of this model as an avenue for governments, industries, and civil society to develop strategies for waste management and emission reductions. Lowe and Warren [8] added that "the components of this approach include the green design of infrastructure and park facilities (new or modernized), cleaner production, pollution prevention, energy efficiency, and partnership between companies. An EIP also seeks benefits for neighboring communities to ensure that the impact net of its development is positive". In turn,

some disadvantages of the EIPs are the complications of management, relations between agents, and compliance with regulatory requirements.

Since their inception, EIPs have attracted the keen interest of industrial planners worldwide, and today, they are widely considered to be a viable and worthwhile alternative to traditional industrial parks. Hundreds of EIPs have been developed in Japan, Australia, Canada, etc. Indeed, since the term "EIPs" first appeared in the early 1990s [16], different experiments have tested the approach—first in North America and later on in Europe, Asia, and South Africa, leading to several practices of interest [17].

These approaches, based mainly on business-to-business exchanges, are only a few of the possibilities of what an EIP could look like. Other planning strategies have also been proposed, for example, the China Circular Economy (CE) in 1998. This model incorporates cleaner production methods and industrial ecology as part of a larger system encompassing industrial enterprises [18]. Ultimately, the EIP framework envisions an ecological planning approach for industrial areas [19] that moves beyond the idea of merely controlling pollution to encourage "thinking as an ecosystem" [20].

While industrial ecology is still a nascent field, it offers a promising approach for achieving environmentally sustainable economic development. It also reflects the wide acceptance—heralded by the 1980 launch of the World Conservation Strategy, the 1987 report of the World Commission on Environment and Development Our Common Future, and the 1992 Agenda [21]—that environmental integrity, economic efficiency, and social equity are all key components of sustainability.

More recent approaches to ecological planning incorporate principles of landscape ecology, which focus on implementing EIPs within a spatial dimension [20]. The advent of industrial ecology in the late 1980s added a new perspective to industrial development [22–24], establishing the goal of designing industrial complexes that mimic natural ecosystem processes [25].

The basis for EIPs is rooted in industrial ecology, which was first described by Frosch and Gallopoulos in 1989 [16]. According to these authors, an industrial ecosystem consists of a system in which "the consumption of energy and materials is optimized, and the generation of waste is minimized as the effluents of a process". The symbiotic nature of EIPs improves the efficiency of the industrial processes, as the collective benefits (i.e., the benefits obtained by individual companies) exceed the sum of the parts.

B.H. Roberts has explored EIPs in Australia [15], one of the highest waste producers in the world, where waste has—until very recently—been considered of little use and less value. In the industrial sector, waste has typically been perceived as nothing more than a costly by-product of manufacturing. However, as EIPs can channel the industrial ecology approach, they have attracted the interest of governments, companies, and citizens who are looking for ways to scale up waste recycling and reduce emissions.

### 1.2. Methodologies for the Sustainability Assessment of Urban Solutions

The magnitude of the environmental impacts caused by the implementation of engineering projects and productive activities throughout their life cycles has been measured with different tools and indicators. Peuportier [26] carried out an extensive presentation of environmental indicators within the urban environment, proposing eco-techniques for efficient construction that take into account energy optimization and the use of renewable energies. Other authors, such as Jullien [27], have analyzed eco-design for infrastructure, using life cycle analysis (LCA) tools and calculating the environmental impact of solutions using ECORCE, a French program for reducing energy consumption and emissions. This author has also examined eco-indicators that quantify the potential impact of energy consumption, global warming, acidification of the environment, alteration of ozone levels, eutrophication, and human toxicity. All this research analyzes different transport systems (rail, road, etc.) from the perspective of implementation of the infrastructures. Currently, there are satisfactory methodologies at the international level. For example, BREEAM [28] is a sustainability assessment method for the master planning of projects, infrastructure, and buildings, while CASBEE

(Comprehensive Assessment System for Built Environment Efficiency) [29] is a green building management system in Japan, applicable on different scales. from the building to the city. Other examples include LEED (Leadership in Energy and Environmental Design) [30], which uses a green building rating system, and—at the national level—SPRILUR [31], which is the certification guide for ecological urban planning [32].

These systems enable the evaluation of different urban planning activities, granting distinctions or values in compliance scales that consider different attributes, such as the landscape, land use, the ecosystem, risk, and the life cycle. Additionally, there are other tools, such as SIMAPRO LCA software [33], for evaluating the environmental impact of industrial products during the entire life cycles of construction processes.

Constructing a section of road prompts the development of infrastructure inherent to any urban fabric, including longitudinal sections, construction elements, facilities, and other elements of infrastructure. This constructed road section is a civil engineering design solution for specific functional needs, but it incurs an environmental impact that can be evaluated and minimized, especially during the phase of material choice and system construction. Nowadays, the common elements of the road cross-sections that characterize urban plots in residential and industrial areas are:

- Road traffic lines
- Sidewalks
- Median strips
- Berms
- Parking bands attached to the road
- Special lines or carriageways

The longitudinal facilities of each one of the infrastructure services that normally run through the subsoil are represented as so-called prisms, facility bulbs or, alternatively, service galleries. Figure 1 shows a schematic representation of a road cross-section, with a clear illustration of the elements within it.

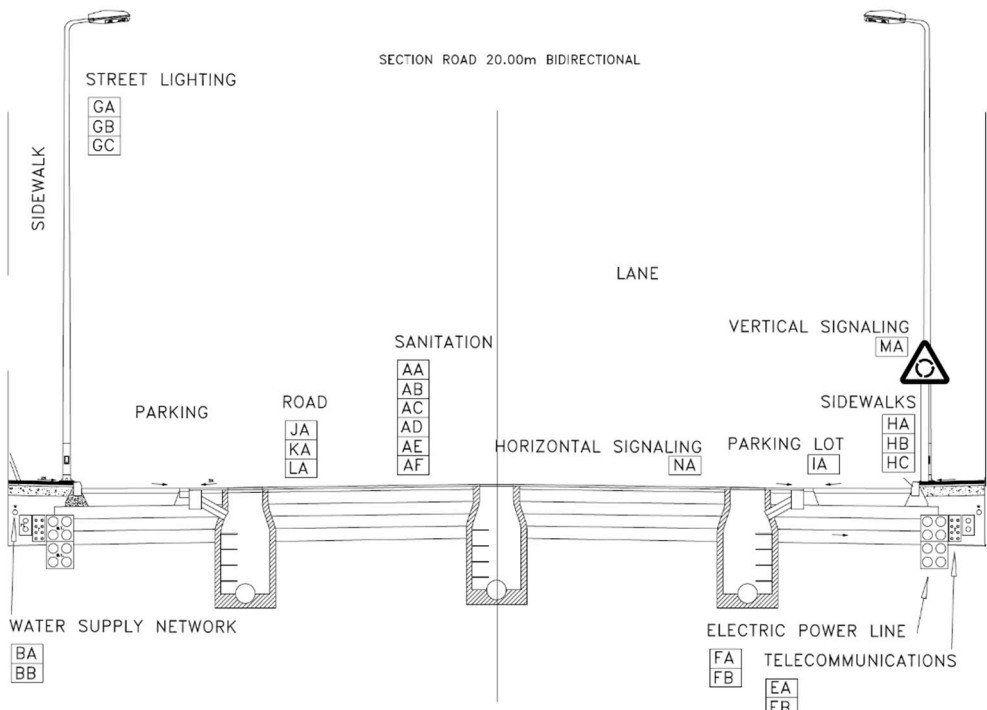

**Figure 1.** Road section, including all identified variables.

This paper proposes an optimization methodology to support decision-makers, that is, local authorities and/or industrial park operators, during the EIP deployment process. An array of commercial possibilities for infrastructure, such as sanitation, water supply, energy distribution, gas distribution, telecommunications networks, and the roadway structure, are available to choose from.

In this study, different parameters have been chosen for each solution, linking each to three criteria used to justify the optimization tool solutions. These parameters are the economic cost (€/m$^2$), the amount of embedded energy (MJ/m$^2$), and the amount of $CO_2$ emitted per m of the proposed solution.

The latest growth in heuristic optimization techniques is linked to artificial intelligence procedures. In multi-objective problems, the objectives are generally in conflict. For example, for the construction of a water dam, an electricity provider is interested in maximizing storage capacity, while at the same time minimizing construction costs and water loss due to evaporation. The company must decide on the man-months used for construction as well as the mean radius of the lake, and it must also respect certain constraints, such as the minimum strength of the dam. Here, the set of alternatives (possible dam designs) is infinite. The criteria are functions of the decision variables to be maximized or minimized, and they are clearly in conflict: For instance, a dam with a large storage capacity is incompatible with a low construction cost [34].

For the constructed road section, this is a real engineering problem, whose solution should aim to minimize economic cost (€/m), $CO_2$ emissions (kg$CO_2$/m), and the embedded energy in the solution (MJ/m). These problems are both difficult and realistic. Genetic algorithms (GA) are widely used meta-heuristics that are very apt for dealing with this kind of problem. GAs are commonly tailored to accommodate multi-objective problems, with special fitness functions and other methods to promote solution diversity. The multi-objective optimization used is the NSGA-II algorithm [35], where the objectives are carbon dioxide emissions for the construction of 1 m of a road of 20 m, the economic cost (€/m), and the embedded energy (MJ/m). The NSGA-II algorithm outperformed others in the Pareto-based optimization process. NSGA was designed for, and is suited to, continuous-function multiple-objective optimization problem instances. The non-dominated sorting genetic algorithm is a multiple objective optimization (MOO) algorithm and is an example of an evolutionary algorithm from the field of evolutionary computation. There are two versions of the algorithm, the traditional NSGA and the updated and currently canonical form, NSGA-II.

## 2. Optimization Problem Definition: Geometry and Materials

The multi-objective optimization problem for the road section entails an economic and sustainable optimization, which involves the minimization of different objective functions:

$$\min F(x) = [F_1(x),\ F_2(x), \ldots, F_k(x)], \tag{1}$$

where $F(x)$ is an array of objective functions and $x = [x_1,\ x_2,\ \ldots,\ x_n]$ is the array of decision variables. Moreover,

$$x_i \in D_i = \left[d_{i1}, d_{i2}, \ldots, d_{iq_i}\right],$$

where $D_i$ is the set of values $d_{ij}$ that can be adopted by the $i$-variable $x_i$ of the array $x$ (discrete variable domain).

### 2.1. Problem Variables

All objective functions ($F_i$) depend on variable ($x$) vectors. Moreover, the aim of this discrete variable optimization problem is to identify a feasible solution that generates the lowest value for the objective functions.

The design variables ($x$) consist of the magnitudes subject to variation during the optimization procedure. Their influence on the optimal design is studied, and the best values are identified. Specifically, 26 variables were established to define the road cross-section. These variables can be classified into different groups, which include road infrastructure, such as the road base and the

sidewalk surface, and also important utilities, such as water and electrical supplies. A complete list of all variable definitions is included in Table 1. This exhaustive list was defined, taking into account all infrastructures that may affect the road cross-section definition. Besides this, the table shows (column Var.) the number of different solutions that can be adopted for each variable.

**Table 1.** Problem variables.

| Id. | Var. | Group | Description |
|-----|------|-------|-------------|
| AA | 11 | | Pipe material |
| AB | 6 | | Registration well |
| AC | 7 | | Chest |
| AD | 7 | Sanitation | Scupper |
| AE | 2 | | Chest cover |
| AF | 6 | | Filling |
| BA | 6 | Water supply | Pipe material |
| BB | 6 | | Filling |
| CA | 5 | Fire protection | Pipe material |
| DA | 2 | Gas supply | Pipe material |
| EA | 4 | Telecommunications | Plumbing prism |
| EB | 7 | | Cabling |
| FA | 4 | Electric supply | Canalization |
| FB | 11 | | Cabling |
| GA | 8 | | Crosier |
| GB | 3 | Street lighting | Luminary |
| GC | 5 | | Lighting group |
| HA | 9 | | Surface |
| HB | 3 | Sidewalks/footpath | Curb |
| HC | 4 | | Curb base |
| IA | 7 | Parking lot | Material |
| JA | 8 | | Esplanade |
| KA | 4 | Road way | Base |
| LA | 14 | | Firm |
| MA | 8 | Signaling | Signaling element |
| NA | 5 | Road paint | Road paint |

A specific configuration of the defined design problem is shown in Figure 1. This represents a solution, where all variables are assigned within the feasible range of variability. For instance, PVC, PE, or PP are available for pipe material, fiber-reinforced concrete, asphalt, and recycling aggregates for the roadway firm. The selection of each one of these elements may affect the final solution in terms of economic cost and sustainability.

Working within the variability defined for each variable, and due to the combinatorial explosion, the extension of the solution space can be obtained as follows:

$$\prod_{i=1}^{26} q_i = 11 \times 6 \cdot \ldots \cdot 8 \times 5 = 2.798 \times 10^{19}.$$

With the current available computational capacity, 330 solutions can be processed per second [36]. This means that the estimated time for the exhaustive exploration of the entire solution space reaches $8.4 \times 10^{16}$ s, or $20 \times 10^6$ centuries. An alternative approach is thus clearly needed to achieve an optimal solution to the problem, with an acceptable computational time. Approximation algorithms (e.g., genetic algorithms) are a good alternative approach to these combinatorial optimization problems. The specific genetic algorithm used for this problem is described in the following section.

## 2.2. Objective Functions

As explained above, the multi-objective optimization problems involve the analysis of different objective functions to be minimized. In this case, three objective functions were defined:

- Economic cost, ($E(x)$)
- Eco-price generation of $CO_2$ in the manufacturing process ($CO_2(x)$)
- Energy consumption, ($MJ(x)$)

The first objective function, economic cost ($E(x)$), is the most typical indicator in this type of optimization analysis. It is a good index for providing information about the amount of resources used in the complete road execution. The indicator unit is €/m, which measures the cost of each road meter.

The second and third objective cost functions are related to the environmental sustainability of the road construction.

Specifically, the $CO_2$ indicator ($CO_2(x)$) measures—in kg of $CO_2$/m—the amount of carbon dioxide emission resulting from the use of materials throughout the entire life cycle of road construction. Material emissions are evaluated at different stages of construction, transport, and even demolition. Finally, the energy function ($MJ(x)$) quantifies the total energy used, in MJ/m, for each element's manufacture, transport, and installation.

In order to obtain the value of each indicator for the elements selected in the road definition, we used the Bank Structured Data of Building Elements (BEDEC) [37]. In the Spanish context, this data bank was created by the Institute of Technology Construction of Catalonia (ITeC).

The ITeC BEDEC is a parametric bank that contains more than $50 \times 10^3$ items used in new construction and maintenance for buildings, urbanization, civil engineering, and elsewhere. It contains reference prices in the Spanish market for more than 93 companies, and includes environmental data such as construction waste and packaging, energy cost, and $CO_2$ emissions. Using this database may reduce the scope of analysis to the Spanish context. However, the procedure followed to solve this optimization problem can be used for other world regions, substituting the database used for unit prices.

Finally, the magnitude of each objective function can be obtained by adding the unit indicator of each element:

$$F(x) = \begin{cases} F_1(x) = E(x) = \sum_{i=1}^{n} E(x_i) \\ F_2(x) = CO_2(x) = \sum_{i=1}^{n} CO_2(x_i) \\ F_3(x) = MJ(x) = \sum_{i=1}^{n} MJ(x_i), \end{cases} \tag{2}$$

where $E(x)$ is measured in €/m, $CO_2(x)$ in $KgCO_2$/m, and $MJ(x)$ in MJ/m.

## 3. Methods: Applied Heuristic

In this problem, as described previously, combinatorial explosion means that the analysis of the entire solution space is an impossible task, even with the large computational capacity available nowadays. However, approximation algorithms (e.g., heuristic algorithms) are available to explore the most promising solution space areas and can reach an approximate optimal solution within an acceptable time period. One of the most useful and well-known global optimization techniques in the heuristic group is the genetic algorithm (GA). This robust approach follows principles of evolution; for instance, inheritance, selection, crossover, and mutation. In a GA, chromosomes are used analogously to encode a candidate solution, and the objective (or fitness) functions work as quality indicators for individuals. As the population progressively evolves, its average fitness increases, a derivation of the Darwinian principle of the "survival of the fittest". Generally speaking, the selection of the highest quality genes is stochastic. Selected individuals are operated by means of the process of the genetic operator—for instance, selection and mutation—until the algorithm is concluded. A general GA introduces methods to promote quality and diversity in the solution.

### 3.1. Random Walks

In multi-objective optimization problems, the objective functions are inherently in conflict. As a first step in the heuristic algorithms, a random walk (RW) was carried out. An RW involves repetitive random generation of a solution. In this process, the solution and objective function construction are assessed, and some correlations between objective functions may be found.

Figure 2 shows the results of a $500 \times 10^3$ iteration RW. The objective cost function E ($\text{€}/m$) is independent from $CO_2$ and energy, because there is a low observed correlation in the RW ($R^2 = 0.32$–$0.34$). However, objective sustainability functions ($CO_2$ and energy) show a strong correlation between them ($R^2 = 0.96$). This means that the three objective functions defined are in conflict in pairs (E($\text{€}/m$) vs. $CO_2$ and E($\text{€}/m$) vs. energy).

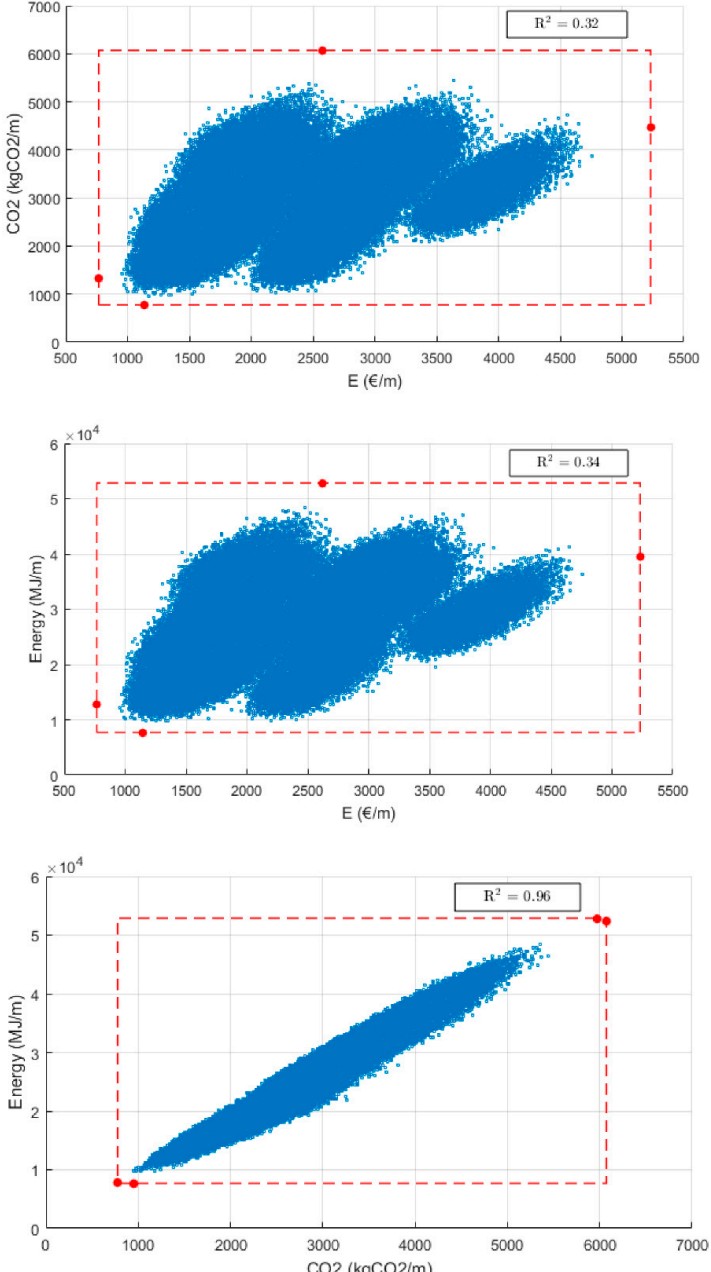

**Figure 2.** Random walk (RW) for the $500 \times 10^3$ iterations.

Additionally, the RW was useful for showing the solution space limits (see Table 2). In Figure 2, a red contour box was also plotted, representing the limit solutions for each objective function. The minimum solutions for each target can be obtained by selecting the best option for each variable. These solutions are plotted as red dots in Figure 2. However, the minimum solution in terms of economic cost (€/m) is not the same as the minimum for $CO_2$ emissions. Therefore, the aim of the heuristic optimization process will be to obtain an optimal solution that simultaneously minimizes all objective functions.

**Table 2.** Objective space limits.

| Id. | Min | Max |
|---|---|---|
| Economic obj. | 766.1 €/m | 5231.3 €/m |
| $CO_2$ | 776.7 kg$CO_2$/m | 6072.2 kg$CO_2$/m |
| Energy | 7700.2 MJ/m | 52,856.0 MJ/m |

### 3.2. The Pareto Set: An Optimal Frontier

Multi-objective optimization is an important research field, because most real problems have more than one objective to be taken into account. The techniques developed to solve these types of problems are called multi-objective combinatorial optimization techniques.

The simplest way to deal with several objectives would be to merge them into a single one using any arithmetic operator. This approach may be straightforward, but it also has important disadvantages—one objective, for example, may dominate the others. To avoid this problem, another technique, called a Pareto-optimal solution, may be applied to multi-objective optimization problems.

Using the Pareto dominance concept, one problem solution ($x_1$) dominates another one ($x_2$) in a multi-objective minimization problem, when:

$$x_1 \prec x_2 \; if \; \forall i \in \{1, \ldots, m\} F_i(x_1) \leq F_i(x_2). \tag{3}$$

The set of solutions not dominated by any other in the solution space ($S$) is called the Pareto optimal set ($P$):

$$P = \{x \in S | \nexists x' \in S : x' \prec x\}. \tag{4}$$

Figure 3 graphically represents the Pareto optimal set for a problem with two objectives to be minimized.

The Pareto optimal set is presented here as the multi-objective optimization problem solution. Therefore, a unique solution is not obtained. Other multi-criteria decision techniques should be applied afterwards to choose a particular solution of the Pareto frontier.

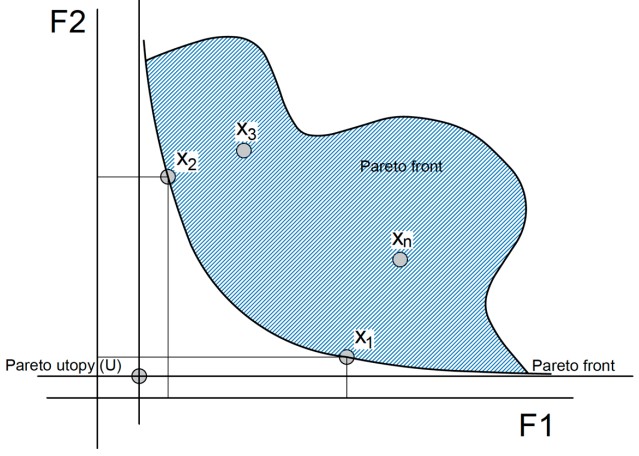

**Figure 3.** Pareto optimal set.

### 3.3. Non-Dominated Sorting Genetic Algorithm (NSGA)-II for Multi-Objective Optimization

We chose a fast non-dominated sorting genetic algorithm (NSGA-II) [35], as a multi-objective genetic algorithm (MOGA), to deal with this problem solution.

MOGA algorithms work on a population (i.e., a set of solutions in the solution space) through a group of operators. The baseline population is generated randomly and thereafter through genetic operators, like crossover and mutation, using the non-dominated rank and a crowding distance.

Each individual is assigned a non-dominated rank according to its relative fitness. A fitness function is a particular type of objective function that is used to summarize, as a single figure of merit, how close a given design solution is to achieving the set aims. Fitness functions are used in genetic programming and genetic algorithms to guide simulations towards optimal design solutions.

The total population is sorted in non-dominated consecutive front, following the Pareto optimal approach. Solutions are sorted into each Pareto front using a crowding distance, which measures the proximity between two population members of the same frontier. The individuals with the largest crowding distance are sorted beforehand to preserve diversity during the performance of the algorithm. While the non-dominated rank maintains the elitism, the crowding distance helps to preserve diversity in the solution. This is very important in order to reach a good convergence, which is important for approximating global optimum solution members.

First, a random parent population, with size $N$, is created ($P_0$). This population is sorted following the rank assignment and the crowding distance, as explained above. Then, using this sorted list, the tournament selection, crossover and mutation are used to create an offspring population ($Q_0$). This offspring population is added to the previous parent one ($P_0 + Q_0$) in order to preserve elite members. Both populations are sorted again together, based on non-domination, and a subset of size $N$ becomes the new parent population ($P_1$). This procedure is applied repetitively, obtaining consecutive parent populations ($P_k$), until it is stabilized, at which point the algorithm stops.

## 4. Optimization Results and Discussion

Figure 4 presents the optimization results obtained through the NSGA-II algorithm, showing that the Pareto set (pink dots) is well-structured between the minimum values of each objective function (red dots). Additionally, the optimization algorithm has shown a good search performance, obtaining optimal points far from the RW results (blue dots) and in the correct minimization direction. Moreover, due to the strong correlation found previously between $CO_2$ and energy objective functions, the Pareto frontier between them is omitted. Only economic (E) and $CO_2$ objective functions will be considered hereafter.

Once the Pareto set has been created, several optimal solutions to the problem are available. In order to select one of them, all objective functions should be combined into a single one. It will be useful to sort all optimal solutions to identify the best. The combination of objective functions may do this by defining weights. In this case, the most common decision involves assigning equal weights to all objective functions. However, in this paper, additional weights have been defined to compare a range of solutions of the Pareto set. Specifically, the weights defined are shown in Table 3.

$$F'(x) = w_E \parallel E(x) \parallel \; + w_{CO_2} \parallel CO_2(x) \parallel \qquad (5)$$

**Table 3.** Objective weights.

| Id. | $w_E$ | $w_{CO2}$ |
|-----|-------|-----------|
| OP-A | 0.5 | 0.5 |
| OP-B | 0.33 | 0.66 |
| OP-C | 0.66 | 0.33 |

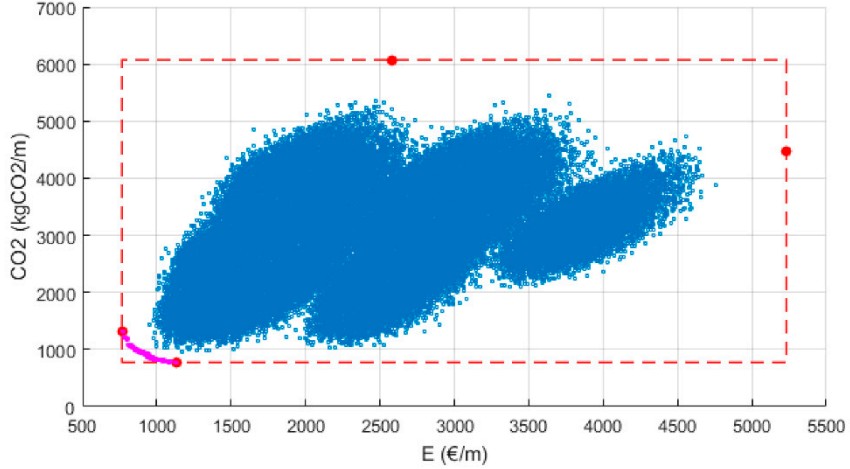

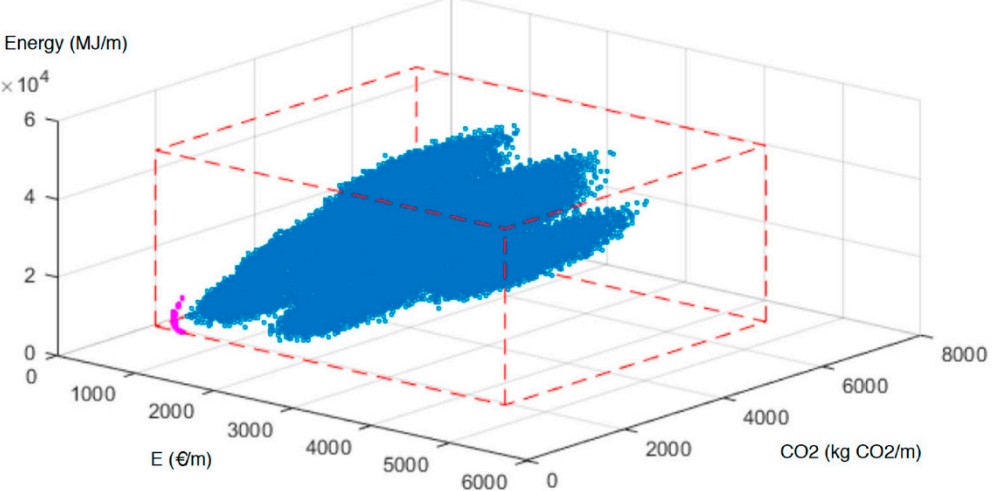

**Figure 4.** NSGA-II Pareto results.

Adding all objective functions together required their prior normalization in order to avoid any objective domination. The normalization of each objective was based on the objective range from Table 2.

Finally, all Pareto set solutions can be projected according to the selected objective function weights (see Figure 5). For instance, the projection of the Pareto set in the bisector line (red dots) allows us to sort optimal points using equal weights. Therefore, three sorted lists can be obtained from the weights defined above, obtaining the best solution for each option.

Using the selection methodology described previously, three different solutions of the Pareto frontier were finally selected.

Figure 6 shows, through graphic patterns, the resulting elements from the full range of possibilities that make up the optimal genetic combination (one of the possibilities) in relation to the sustainability of the solution. The graphic representation by would be modified, means of frames, if any of the components of the constructive solution were altered.

Table 4 describes the results of the optimal solution OP-A (equal weights) for each problem variable set out in Table 1. The optimal solution was thus completely defined. Its economic cost is 943.7 €/m, the $CO_2$ indicator is 855.3 kg$CO_2$/m, and the energy consumption is 8606.3 MJ/m.

Furthermore, OP-B and OP-C weight configurations were also processed, with very similar solutions. In fact, Tables 5 and 6 show only the variation of these optimal solutions, compared with OP-A. All variables not listed in Tables 5 and 6 adopt the same value as that in Table 4.

Only four variables change from OP-A to OP-B or OP-C. This shows the robustness of the optimal solution obtained.

In the optimal solution OP-B, a higher weight for the $CO_2$ indicator was established, resulting in an improvement in the sustainability indicators (798.2 kg$CO_2$/m and 8005.8 MJ/m), albeit at a slightly higher economic cost. On the other hand, OP-C offers a cheaper solution, with a moderate worsening of sustainability indicators. These variations can be reached through small changes in only four problem variables.

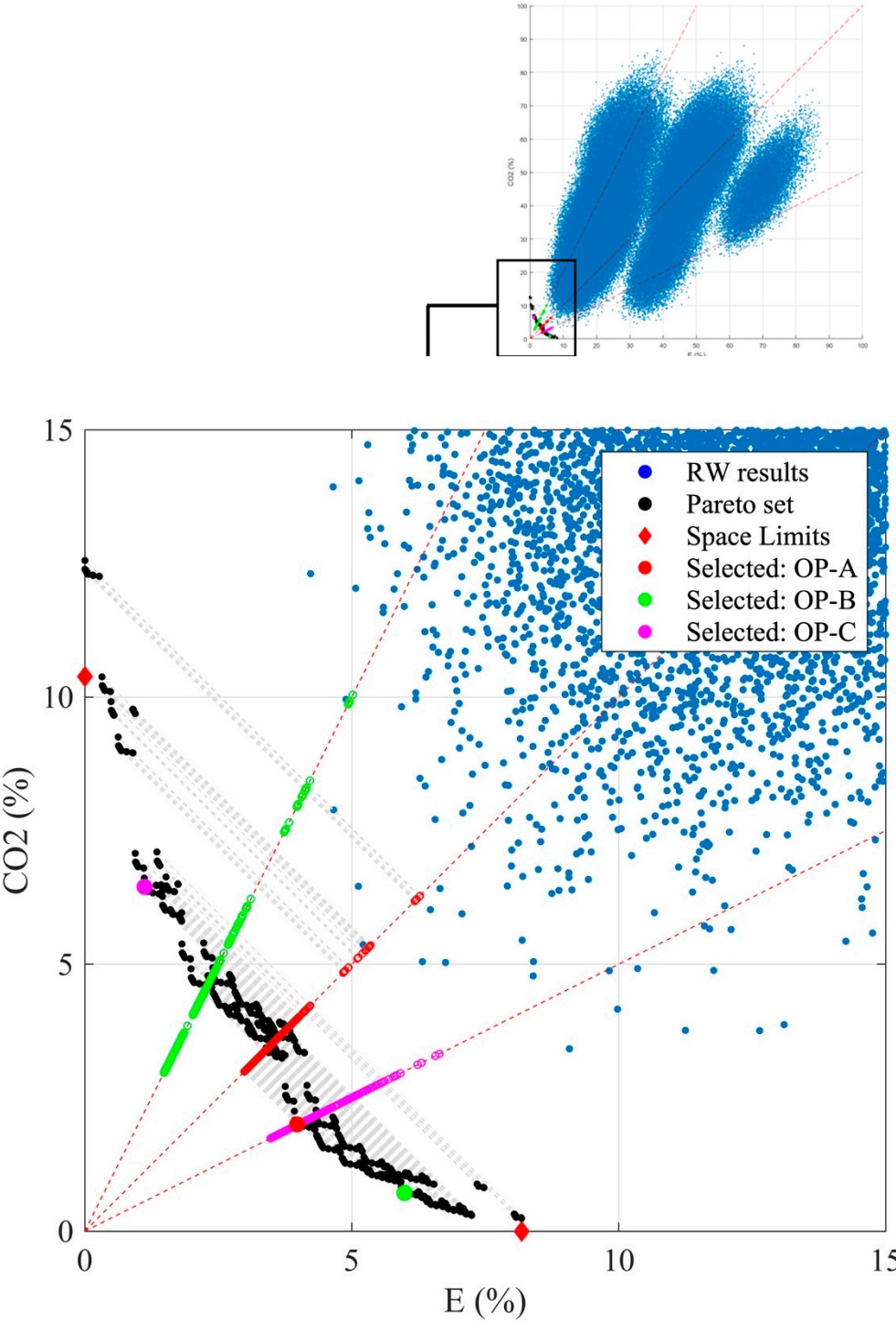

**Figure 5.** Pareto results selection.

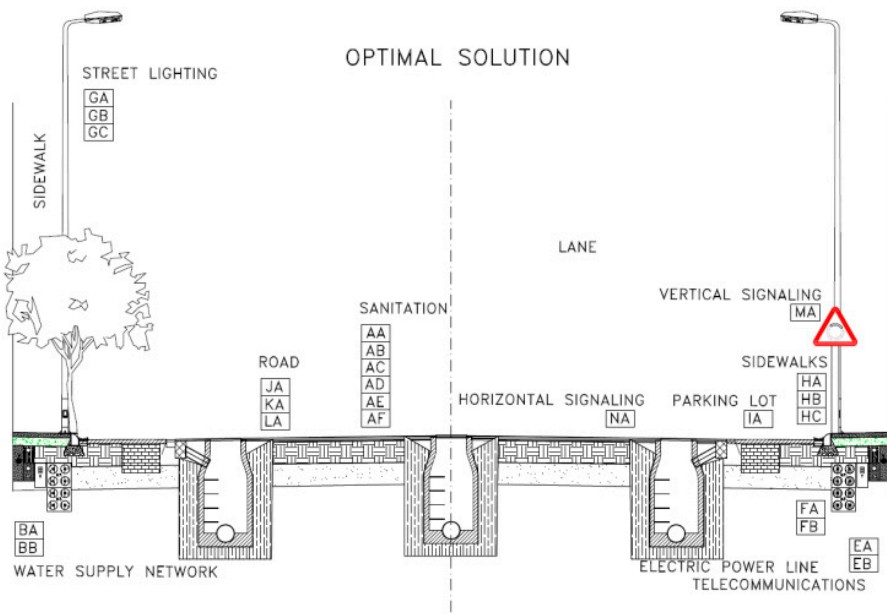

**Figure 6.** Optimal solution.

**Table 4.** Optimal solution OP-A.

| Gene | Allele | Phenotype | % € | % CO$_2$ | % E |
|------|--------|-----------|-----|----------|-----|
| AA | 3 | Polyethylene pipe | 6.81 | 3.80 | 2.49 |
| AB | 6 | Polypropylene log well | 4.35 | 9.39 | 8.28 |
| AC | 2 | Brick register check | 5.72 | 5.88 | 5.95 |
| AD | 5 | Polyethylene scupper | 1.82 | 2.97 | 2.68 |
| AE | 1 | Reinforced concrete cover | 7.97 | 19.2 | 13.2 |
| AF | 6 | Filling own lands | 1.82 | 4.12 | 4.36 |
| BA | 6 | Ethylene–propylene pipe | 1.00 | 0.67 | 0.45 |
| BB | 6 | Filling own lands | 1.82 | 4.12 | 4.36 |
| CA | 1 | Polyethylene pipe | 0.57 | 0.36 | 0.23 |
| DA | 1 | Polyethylene pipe | 0.35 | 0.19 | 0.13 |
| EA | 4 | PVC pipe | 0.33 | 0.18 | 0.12 |
| EB | 1 | Copper conductor | 0.13 | 0.08 | 0.07 |
| FA | 1 | PVC pipe | 0.33 | 0.18 | 0.12 |
| FB | 4 | Copper conductor | 0.09 | 0.16 | 0.11 |
| GA | 1 | Column polyester fiberglass | 22.7 | 7.15 | 6.97 |
| GB | 2 | Aluminum luminaire | 15.0 | 6.04 | 4.07 |
| GC | 5 | LED lamp | 1.59 | 0.98 | 0.92 |
| HA | 5 | Reinforced concrete | 1.89 | 5.76 | 5.65 |
| HB | 3 | Prefabricated concrete parts | 2.18 | 1.39 | 1.42 |
| HC | 1 | Prefabricated concrete parts | 0.80 | 1.89 | 1.94 |
| IA | 7 | National granite blocks | 5.98 | 2.90 | 2.75 |
| JA | 1 | Tolerable soil | 1.23 | 3.21 | 3.12 |
| KA | 3 | Macadam | 5.97 | 7.03 | 11.4 |
| LA | 4 | Bituminous mixture | 6.63 | 9.87 | 16.8 |
| MA | 6 | Wooden post | 1.85 | 1.94 | 1.89 |
| NA | 3 | Polyurethane paint | 1.06 | 0.57 | 0.53 |
| E = 943.7 €/m; CO$_2$ = 855.3 kgCO$_2$/m; energy = 8606.3 MJ/m | | | | | |

**Table 5.** Optimal solution OP-B (variation from A).

| Gene | Allele | Phenotype |
|------|--------|-----------|
| AC | 5 | PE register check |
| FB | 9 | Aluminum conductor |
| HA | 9 | Cobblestones |
| HC | 3 | Stone blocks |
| | | E = 1033.5 €/m; $CO_2$ = 798.2 kg$CO_2$/m; energy = 8005.8 MJ/m |

**Table 6.** Optimal solution OP-C (variation from A).

| Gene | Allele | Phenotype |
|------|--------|-----------|
| EA | 3 | Corrugated polyethylene pipe |
| FA | 2 | Polyethylene pipe |
| GB | 3 | Polypropylene luminaire |
| IA | 6 | Concrete paver |
| | | E = 816.2 €/m; $CO_2$ = 1054.1 kg$CO_2$/m; energy = 10,163.4 MJ/m |

## 5. Conclusions

A novel method for selecting sustainable solutions in the construction process of roads and infrastructure was proposed and shown to provide minimal environmental costs, based on three parameters: $CO_2$ emissions, embedded energy, and economic cost. The NSGA-II algorithm can select individuals that are represented by a flat Euclidean space, which satisfies the position requirements using an elite conservation strategy, an adaptive mechanism, a hybrid genetic algorithm, and reasonable designs of the fitness function, plus selection, crossover, and mutation operators. Comprehensive simulations were conducted, demonstrating that the NSGA-II-based sustainable solution method can effectively select an optimal subset from the available individuals.

The results of the sustainability analysis for the case study in this real engineering problem are presented. First, the effects of environmental factors on the total construction cost of the model are compared with the results from the models, including sustainability considerations. The optimal solution associated with the road section was identified from a large number of possibilities using a computer algorithm based on genetic algorithms, with individuals defined by carbon footprint, embedded energy, and economic cost. In the case study, it was possible to find a constructive solution, with a $CO_2$ emissions level and embedded energy level that were 40% less than those of conventional solutions, and a reduction in the economic cost of up to 30%. These variables were added to the model in order to analyze the impact of the system on the environment. Then, the model was solved by generating individuals, crossing them, and generating mutations and functions by selecting them based on their suitability, legal compliance, and other considerations. The optimal individuals were those that were placed on a three-dimensional plane, Pareto, with the shortest distance to the origin of coordinates.

Presenting the model together with the algorithm chosen for the search for an optimal solution shows the immense potential of this heuristic technique in the resolution of complex construction problems, where the criteria of choice are multi-objective, based on a combinatorial problem of a large field of possibilities within each minimizable objective pursued. The minimum value for an objective is usually associated with non-optimal values for another. If several objectives intervene simultaneously, the solution is not trivial. For example, the construction of a bridge or a dam, where the objectives to be minimized are $CO_2$ emissions, the amount of structural steel, the type of concrete, time, etc., or a complex building, whose optimizable parameters are the economic cost, $CO_2$ emissions, waste generation, and time.

A specific conclusion of this work can be expressed as follows. In the sustainable optimization design of steel and concrete structures, based on $CO_2$ emissions, it can be concluded that the optimal sustainable solution is very close to the optimal economic one. This can be explained by the fact that, in

this case, the objective functions, economic and sustainability, do not contradict one another. Therefore, the problem can be modelled as a single objective optimization problem. However, the previous conclusion does not work with the problem of the urban design optimization. In this problem, multiple materials and processes should be taken into account together, and the optimal sustainable solution differs from the economic one. Thus, the problem must be always modelled as a multi-objective problem to obtain the Pareto set, economic-$kgCO_2$, of the efficient solutions. Additionally, the selection of the solutions from the Pareto set requires the development of a multi-criteria problem.

Within the field of solutions, the minimums and maximums of each objective are perfectly known in advance. However, the minimum is not evident, since it is necessary that the resulting individual is technically viable and complies with legal, technical, and design restrictions, which requires a study of the said solution, so that its suitability can be assured. The need to use the chosen heuristic method is motivated by the fact that the immense population field to explore would impose excessive and unacceptable temporary needs to provide an efficient solution.

A genetic algorithm, and within it, the NSGA-II method, is demonstrated as a powerful tool for the resolution of a multicriterial combinatorial problem.

**Author Contributions:** S.M.R. has performed this research with the collaboration of C.G., A.H., and V.A. All authors read and approved the final manuscript.

**Funding:** This research received no external funding.

**Conflicts of Interest:** The authors declare no conflict of interest.

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
