# Peer review of "Sustainability Assessment of Constructive Solutions for Urban Spain: A Multi-Objective Combinatorial Optimization Problem"

_sustainability, doi:10.3390/su11030839_

Round 1

Reviewer 1 Report

The manuscript titled “Sustainability assessment of constructive solutions for urban Spain. A multiobjective combinatorial optimization problem” is a resubmitted paper. It has been substantially improved through revision when compared to its previous version. Most of my questions and comments have been properly resolved. I listed 2 additional comments below, and I believe the manuscript can be accepted after the authors solve or justify them.

1. Figure 6: This figure was not mentioned throughout the paper. While it is quite similar to Figure 1, what are the differences between these 2 figures? Is it really necessary to have Figure 6 here?

2. Again, the axis titles in Figure 4 are still missing.

Author Response

The reviewer response is in the attached file

Reviewer 2 Report

My comments relate to the paper overall, and especially its context, rather than to technicalities of the method/analysis.

Overall I found this interesting and informative,  and generally appropriately structured and written.  However I would suggest some amendments which are overall relatively minor / easy to accommodate.

Overall: some citations might need checking, as the relevance of a few to the text they support seems unclear (eg 5,6,7 are on cooling and urban trees not green areas / regeneration?  10 seems to relate to Le Corbusier and there are many more relevant sources for other pioneers mentioned; 11 does not seem the best source to refer to details of the location model).

The Abstract is slightly long, and the first sentences are too generic.  Focus on what this paper sets out to do, how it was done, what new knowledge was found. Use of "must" (line 17) should be avoided.

Keywords: "sustainability assessment" better than "sustainability"?

Introduction: section 1 might be improved by sub-sections: at present structure seems a little confused.  Odd to use Brundtland definition of sustainability in abstract but not main text.

p1 para 2: rather sweeping statements need qualifying. "some ... initiatives" (not all); "generally results" etc.

p2 line 1 sense unclear. Presumably it is not that planners consider the construction of green areas unsustainable - construction on green areas perhaps?

p2 line 48 EIPs need defining, ideally where first mentioned

p2 line 66 when did the EIP model respond to these challenges?

p2 para 6 is rather descriptive.  Are there downsides to EIPs?

p2 line 84, "widely considered" - evidence? How many are there?

p2 para 8 would probably be better in a section on defining EIPs

p3 line 106 "adds value to them" - who/what?

p3 para 4 point about acceptance of sustainability (or is it actually sustainable development?) would be better much earlier.

p3 line 137 "urban planning activities" - this is surely broader than just "planning"?

p6 line 229, what current available computational capacity?

p16 conclusions: this is a very short section; there seems to be no discussion of results (but see line 435).  It would be helpful for readers for discussion and conclusions to be separated and expanded.

References: check house style, consistency and completeness of references (eg 18 & 22 lacks year, publication details; 31 lacks access date).

Author Response

(The authors gave the same response as above.)
